# Breaking Earth's shell into a global plate network

C. A. Tang [1,2], A. A. G. Webb [3✉], W. B. Moore [4,5], Y. Y. Wang [6], T. H. Ma [1,6] & T. T. Chen [7]

The initiation mechanism of Earth's plate tectonic cooling system remains uncertain. A growing consensus suggests that multi-plate tectonics was preceded by cooling through a single-plate lithosphere, but models for how this lithosphere was first broken into plates have not converged on a mechanism or a typical early plate scale. A commonality among prior efforts is the use of continuum mechanics approximations to evaluate this solid mechanics problem. Here we use 3D spherical shell models to demonstrate a self-organized fracture mechanism analogous to thermal expansion-driven lithospheric uplift, in which globe-spanning rifting occurs as a consequence of horizontal extension. Resultant fracture spacing is a function of lithospheric thickness and rheology, wherein geometrically-regular, polygonal-shaped tessellation is an energetically favored solution because it minimizes total crack length. Therefore, warming of the early lithosphere itself—as anticipated by previous studies—should lead to failure, propagating fractures, and the conditions necessary for the onset of multi-plate tectonics.

[1] State Key Laboratory of Coastal and Offshore Engineering, Dalian University of Technology, 116024 Dalian, China. [2] State Key Laboratory of Geological Processes and Mineral Resources, China University of Geosciences, 430074 Wuhan, China. [3] Division of Earth and Planetary Science and Laboratory for Space Research, University of Hong Kong, Pokfulam Road, 999077 Hong Kong, China. [4] Department of Atmospheric and Planetary Sciences, Hampton University, Hampton, VA 23668, USA. [5] National Institute of Aerospace, Hampton, VA 23666, USA. [6] Deep Underground Engineering Research Center, Dalian University of Technology, 116024 Dalian, China. [7] School of Resources and Civil Engineering, Northeastern University, 110819 Shenyang, China. ✉email: aagwebb@hku.hk

Fifty years after the foundational works establishing the geometries, kinematics, and mechanics of plate tectonics, we still lack a consensus understanding of how the plate tectonic system initiated[1–9]. Most models envisage initial conditions of a stagnant lid (i.e., a single-plate lithosphere) atop a mantle which was hotter by a few hundred degrees than the present mantle (e.g., refs. [10,11]), such that the key problem would be determining how this lid was first broken. Proffered models suggest that the strength of the lid was overcome by mantle convective forcing (e.g., refs. [7–9]), potentially along locally pre-weakened zones (e.g., ref. [4]); lithospheric gravitational instabilities between oceanic lithosphere and either adjacent oceanic plateau lithosphere[2,6] or adjacent overthickened (i.e., gravitationally collapsing) continental lithosphere[5]; or one or more large bolides[1].

Plate tectonics is the result of a complex series of interactions at the grain, lithologic unit, plate, mantle, and global scales. The geodynamical modeling community has been able to generate plate-like behavior using a number of different fluid-mechanics approaches, including plasticity, damage, and grain-size evolution methods [see ref. [12] for a recent review]. These are based on various degrees of abstraction of the underlying processes that cause rocks to fail and of how this failure modifies the modeled viscous rheology. Despite considerable theoretical effort (e.g., refs. [13,14]), we are unable to directly connect the parameters of these models to the laboratory-derived or microphysical properties of rocks and instead must tune the model parameters to match observations of Earth's plates (e.g., ref. [15]). It is appropriate, then, to bring to bear a new solid mechanics-based approach to the problem of the origin of plate tectonics and the processes by which plate boundaries are initiated.

Here, we use three-dimensional (3D) spherical shell models of a brittle lithosphere to explore the geodynamical processes that initiate tectonic boundaries. We find that thermal expansion in response to anticipated warming of the early lithosphere[3] leads to fracture propagation and eventual coalescence of a global fracture network. Fractures themselves could concentrate volcanism by providing distinctively easy routes to the surface, which in turn could weight the lithosphere locally and thus provide the initial impetus for subduction tectonics.

## Results

**Solid mechanics modeling.** The modeling uses the 3D finite element code RFPA (Rock Failure Process Analysis code—see Methods)[16,17] utilizing up to 3.45 million elements. The models are subjected to quasi-static, slowly increasing interior pressure in a displacement-controlled manner (e.g., induced by gradual thermal expansion, discussed below). Brittle failure is implemented through a strength criterion that represents a stress limit at which the strength drops and fracture occurs. To account for local randomness, each element is assigned a failure threshold obtained from a Weibull probability distribution[16] which contains a homogeneity index, a parameter describing the degree of material homogeneity. Additional material properties (e.g., Young's modulus, failure strength, Poisson's ration, etc.) match the central range of standard rock properties derived from laboratory experiments (see Methods for details; values are listed in Supplementary Table 1). Nine experiments were conducted across model spheres with 6000 km radii. Models 1 through 7 explore different values for shell thickness (20, 40, 60, 80, 100, 120, and 200 km) in order to explore the thickness effect on the fracture pattern. Because we expect that the integrated strength of the lithosphere (i.e., thickness multiplied by modulus) should generally control fracture initiation and growth, we validate our approach by obtaining similar fracture density results by modeling a thin shell (20 km) with a high elastic modulus (250,000

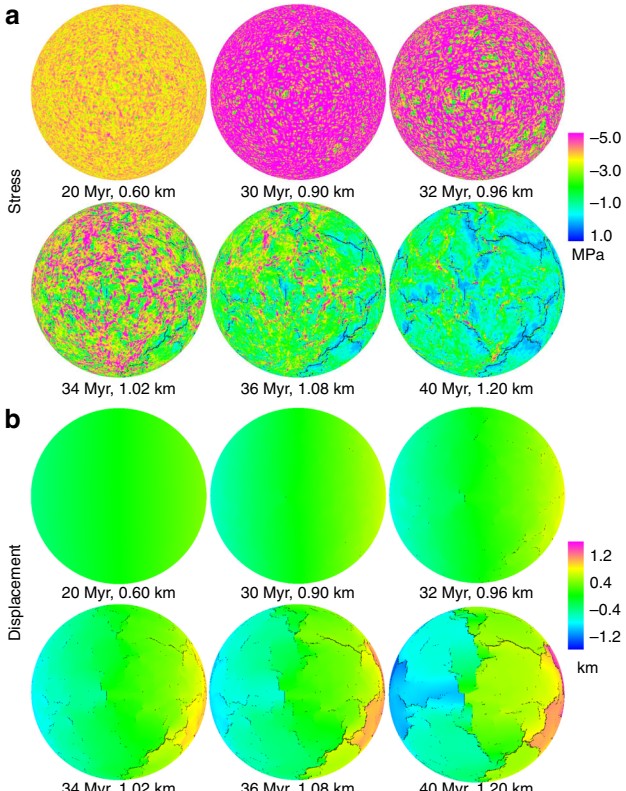

**Fig. 1 Plate-breaking reference model.** The reference model shows the progressive development of a global fracture network in response to displacement-controlled increasing interior pressure (at 0.03 km per million years (Myr)) applied to an 80-km-thick lithosphere. **a** These spheres show the global surface distribution of minimum principal stress magnitudes through time (negative values denote tensile stress, positive values denote compressive stress). **b** These spheres show the associated displacement (articulated along a horizontal axis, from left to right through each sphere's center). In both **a** and **b**, surface elements which attain their ultimate tensile strain are displayed as fractures in black. At 20 Myr, the single-plate lithosphere remains macroscopically undamaged. At 30–32 Myr, with sustained interior pressure, cracks initiate and propagate. By 34 Myr, crack coalescence generates features such as triple junctions. By 40 Myr, with the intersection of most major crack systems, the global fracture network is established, dividing the surface of the sphere into plate-like segments. Supplementary Fig. 4 shows this model in greater detail, with sets of 15 spheres extending from 10 to 100 Myr.

MPa) (Model 8) as well as a thick shell (200 km) with a low elastic modulus (25,000 MPa) (Model 9—see Supplementary Fig. 1).

**Reference model.** Figure 1 shows the development of the fracture pattern evolution for Model 4 (80 km shell thickness, see also Supplementary Movies 1–3). We start the simulations with the undamaged shell loaded from the inner boundary with a radial displacement rate of 0.03 km per million years (Myr). As the shell expands, the total stress in the system increases, and fractures tend to nucleate at the weaker sites (30–32 Myr). In the beginning, fractures do not propagate long distances across the surface, but rather move in small steps from one weak site to the next, occasionally meeting another fracture that is moving in a similar fashion (32–34 Myr). After the initiation of a few longer fractures, most new fractures start near existing fractures and propagate away from their parent fracture, approximately at right angles or triple junctions (34–36 Myr). Finally, successive generations of fractures form, mostly joining older fractures and forming an

array of polygons (after 36 Myr), and the model reaches a critical state resulting in the macroscopic "collapse", in terms of highly fluctuated activities and fracturing avalanches of all sizes. This is a common characteristic of systems that display self-organized criticality[18]. Although randomly distributed, the size of each polygonal fractured area is approximately the same with some deviations due to heterogeneities in the shell. After reaching a certain fracture saturation[19], the number of polygons formed in the fractured shell increases at a reduced rate (after 40 Myr, Fig. 2).

**Parameter exploration.** Similar patterns are observed for all models, albeit with fracture growth rate and fracture spacing (polygon area) variations (Fig. 3, see also Supplementary Figs. 2–10 and Supplementary Table 1). Comparison of the simulations shows that with other parameters held equal, fracture spacing increases with increasing shell thickness and required loading times/amounts for rapid global-scale fracture network development are larger for higher shell thickness. Fractures across the

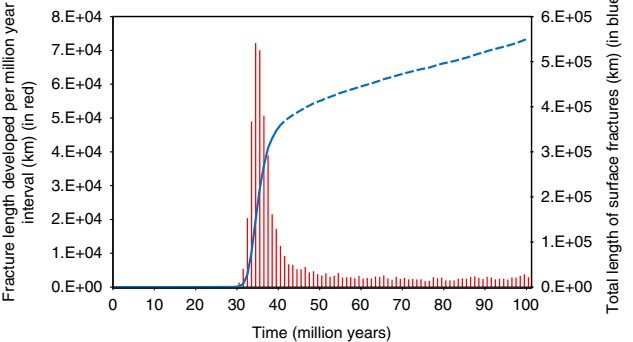

**Fig. 2 Fracture development vs time in the reference model.** Fracture growth rate per million year interval (red), and the integrated length of all fractures (blue), as a function of loading steps in time. Dashed line indicates evolution beyond the thermal expansion limit after 40 Myr (see text).

shell propagate in zig-zag fashion, bending and branching at critical fracture lengths that are controlled by the strength of the shell. As fractures grow and then intersect they become equivalent to plate boundaries which connect at triple junctions (Fig. 1).

## Discussion and conclusions

Our spherical shell models show how a tectonic plate system can evolve from shallow processes. None of the plate boundaries here are actively triggered by deep processes, and a fracture mechanism allows for lithospheric rifting on a global scale. In this scenario, the onset of plate tectonics requires that radial expansion is large enough to induce horizontal stresses that overcome the strength of the lithosphere at some stage in the Archean. With these ingredients, if thermal expansion is sufficient, single proto-rifting zones could initiate and provide the weak zones needed to allow the lid to participate in the convective motions of the interior.

Although nearly all models for early Earth and other hot terrestrial planetary bodies involve a single-plate lithosphere following after a magma ocean, the traditional view has been that early single-plate lithospheric cooling occurred by conduction (e.g., refs. [10,20]), which would produce monotonic lithospheric cooling and thickening through time. However, recent data syntheses and models suggest that early heat loss was dominated by melt advection, partially[21,22] or entirely[3,23,24] through the lithosphere. Models with dominantly volcanic cooling—termed heat-pipe cooling—are consistent with the early Earth's geology[3] as well as the preserved lithospheres of the solar system's other terrestrial planetary bodies[25].

Recently, some researchers[21,22] have argued that intrusive magmatism should dominate early cooling. This argument is largely based on early work[26] which found that Earth's Phanerozoic magmatic systems generally display 1:5 to 1:10 extrusive-to-intrusive ratios. However, such findings were based on limited constraints and on primarily plate tectonic magmatic systems. Although constraints remain limited and substantial variability in extrusive-to-intrusive ratios exists, subsequent work has shown that the basaltic hotspot and basaltic large igneous province

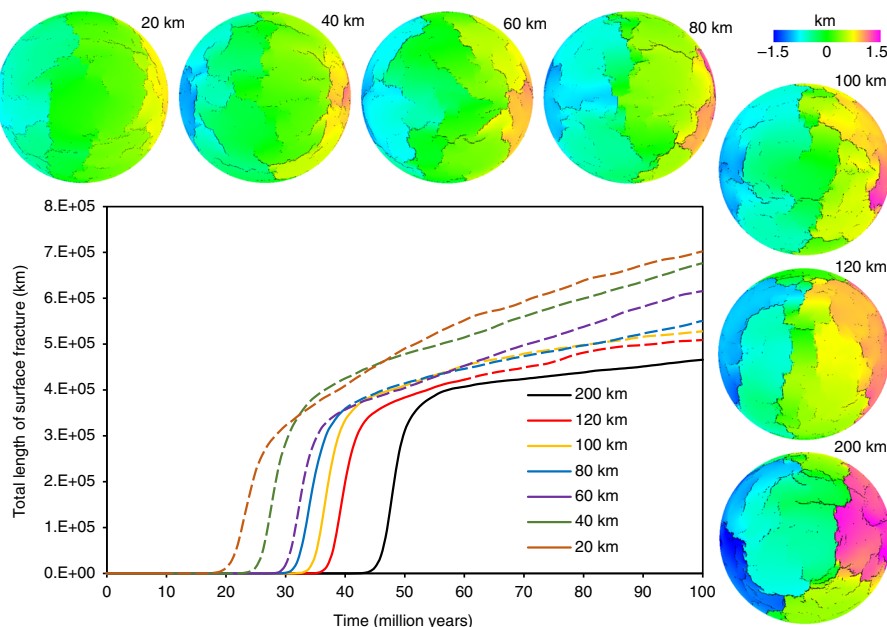

**Fig. 3 Fracture development across shells with various thicknesses.** Surface fracture development for seven model runs with different shell thickness (other variables held constant), imaged in terms of uplift with total fracture lengths. Dashed lines indicate the evolution beyond the thermal expansion limit (see text). The spheres show displacement (articulated as in Fig. 1) as each model run achieves a total length of surface fractures of 4.0E + 05 km.

systems—which are widely accepted as our closest Phanerozoic Earth analogs to the major Archean magmatic systems—have far greater ratios vs all other systems, generally at 1:1 or even 9:1 (refs. [27]–[29]). [Further exploration of this issue can be found in the rounds of reviews and responses to reviewers, published alongside this paper.] In the heat-pipe cooling mode, volcanic advection carries material and heat from the mantle to the surface, where the heat is lost to the atmosphere and the material forms new lithosphere[30]. Continual deposition of melt at the surface produces downwards advection of cold surface material and a thick, cold lithosphere. As heat sources decay, volcanism and downwards advection of cold material wanes, so that conduction increasingly warms and thereby thins the lithosphere[3,25].

The gradual thinning and warming of the lithosphere inherent in the transition out of heat-pipe cooling produces concomitant thermal expansion on the scale required by our numerical models. That is, warming of 50 km of lithosphere by an average of 500° (half the difference between surface and interior temperatures) produces an uplift of 0.75 km (multiplying lid thickness by the thermal expansion coefficient and the temperature change). The thermal conduction timescale for a 50 km lid is about 30 Myr yielding an expansion rate similar to our assumed 0.03 km Myr$^{-1}$ (see Model Validation in Methods below for further discussion on this question). The total amount of uplift from this mechanism depends linearly on the thickness of the lithosphere and therefore ranges from 0.3 to 3 km for lithospheres ranging from 20 to 200 km thick. This limit is indicated by the extent of the solid lines in Figs. 2 and 3 and Supplementary Figs. 2–6 and 11. In practical terms, this means that for our selected parameterization (see Supplementary Table 1), a global rift network develops across lithospheres with thicknesses of 80 km and greater. This assumes that intrusion has not significantly warmed and softened the lid (cf. refs. [21,22]), which would reduce thermal expansion following the end of heat-pipe volcanism and thus potentially limit the development of a fracture network. While the actual ratio of intrusion to extrusion is not well-constrained for any planetary system (both Crisp[26] and White et al.[28] are clear on this point), nor are the processes that control this ratio, it is nonetheless evident that the lithosphere of Io—our solar system's only active heat-pipe body—is not softened by high rates of intrusion, given its support of large topographic features[31].

On Earth, fracturing due to thermal expansion at the end of the heat-pipe era may have produced the initial weaknesses subsequently exploited by plate tectonics. Our numerical experiments show that the polygon fissures could develop on Earth's surface in response to shallow lithospheric processes, with triple junctions as by-products of fracturing. The rapid development of the fracture network in each experiment occurs at around 1 km total expansion and takes ~5 million years, which is similar to the timescales of plate tectonic onset interpreted from the geologies of the Barberton and Pilbara granitoid-greenstone terranes (e.g., ref. [32]). The finding that fracture spacing rises with increasing shell thickness indicates that thicker lithospheres produce larger plates (see also Supplementary Fig. 10)—consistent with what we know about natural fracture spacing in general[19].

The initiation of a global network of rifts, as modeled here, is distinct from plate tectonics because plate tectonics includes creation of new lithosphere at rifts and destruction of old lithosphere at subduction zones. Absent an expanding Earth hypothesis, a global rift network is unsustainable: Earth's surface cannot extend without balancing contraction. However, a host of recent subduction initiation hypotheses are based on the geological development of lateral contrasts in lithospheric buoyancy, with regions of dense lithosphere initiating subduction by sinking into lighter underlying asthenosphere (e.g., refs. [5,6,33]). Similarly, rapid initiation of a global network of rifts might focus volcanism

by providing easy routes to the surface, and thereby weight the lithosphere along the rift shoulders, similar to the volcanic weighting in plume-driven subduction initiation models (e.g., ref. [6]). Portions of this weighted lithosphere could have initiated subduction via the same previously proposed weighting mechanism[2,5,6,33]. The kinematics of this subduction would have thus balanced and sustained rifting along other rift systems, thereby rapidly developing Earth's first global plate tectonic system from a single-plate lithosphere.

## Methods

**Modeling approach.** We use a 3D finite element code, developed by Tang[16] and named RFPA$^{3D}$ (Rock Failure Process Analysis Code), to address the question of how a single-plate lithosphere might fail. The spherical shell models employed here explore a brittle shell layer (Supplementary Fig. 12) that fails under a quasi-static, slowly increasing interior load on the internal boundary of the shell in a displacement control manner. Each model has up to three million elements, with variations primarily dependent on the shell thickness. The simulation proceeds as follows. We increase the load from inside the shell in a displacement control manner. We compute the stresses in each element by solving numerically the finite element method equations. The load is then slowly increased until the stress in some element reaches its threshold. The element is damaged and the disorder is changed according to the rules specified by a probability distribution. We then compute new stresses and repeat the process until no unstable elements are present. Iterating the procedure leads to the final collapse of the models.

The material strength at which a particular element breaks is random, but fixed at the start of the fragmentation process. The Monte Carlo method is used to generate randomly distributed parameters for each element by following the adopted probability distribution of breakdown thresholds[16]. Here we account for this local randomness by assigning to each element a failure threshold defined by the Weibull probability distribution, that is,

$$P(u) = \frac{m}{u_0}\left(\frac{u}{u_0}\right)^{m-1}\exp\left(-\frac{u}{u_0}\right)^m,\qquad(1)$$

where $u$ is the parameter of the element (such as strength or elastic modulus), the scale parameter $u_0$ is related to the average of element parameter, and the parameter $m$ defines the shape of the distribution function. The parameter $m$ defines the degree of material homogeneity and is called the homogeneity index.

Initially, elements are considered to be elastic; their elastic properties are defined by Young's modulus and Poisson's ratio (Supplementary Fig. 13). The stress–strain curve of each element is considered to be linear elastic until the given damage threshold is attained. We choose the maximum tensile stress criterion and the Mohr–Coulomb criterion, respectively, as the damage thresholds. The tensile strain criterion is used primarily to determine whether or not a crack is initiated. If an element is not damaged in tensile mode, then the Mohr–Coulomb criterion is used to judge whether the element damage occurs in shear mode.

When the element is under uniaxial tension, the constitutive relationship is illustrated in Supplementary Fig. 13. It presents an elastic–brittle damage constitutive relation with a given specific residual strength. The stiffness of the elements degrades gradually as the damage progresses, and the elastic modulus of the damaged material can be defined as follows:

$$E = (1 - D)E_0,\qquad(2)$$

where $D$ represents the damage variable, and $E$ and $E_0$ are elastic moduli of the damaged and the undamaged material, respectively.

The damage variable $D$ ranges from zero for the undamaged material to one for damaged material. With regard to the constitutive law that is shown in Supplementary Fig. 13, the parameter $D$ can be calculated as

$$D = \begin{cases} 0 & \varepsilon > \varepsilon_{t0} \\ 1 - \frac{f_{tr}}{E_0\varepsilon} & \varepsilon_{tu} < \varepsilon \le \varepsilon_{t0} \\ 1 & \varepsilon \le \varepsilon_{t0} \end{cases},\qquad(3)$$

where $f_{t0}$ and $\lambda = f_{tr}/f_{t0}$ are the uniaxial tensile strength and the residual strength coefficient, respectively. The residual strength coefficient $\lambda$ is defined as the ratio between the residual strength and initial strength of the element. $\varepsilon_{t0}$ is the strain at the elastic limit, which is the so-called threshold strain for tensile damage, while $\varepsilon_{tu}$ is the ultimate tensile strain, at which the element is completely damaged in tensile mode, so a crack is considered initiate. The ultimate tensile strain is defined by $\varepsilon_{tu} = \eta\varepsilon_{t0}$, where $\eta$ is the ultimate strain coefficient. Herein, $f_{tr}$ is the residual tensile strength, which is given as $f_{tr} = \lambda f_{t0} = \lambda E_0\varepsilon_{t0}$. Then, the above Eq. (3) can be expressed as

$$D = \begin{cases} 0 & \varepsilon > \varepsilon_{t0} \\ 1 - \frac{\lambda\varepsilon_{t0}}{\varepsilon} & \varepsilon_{tu} < \varepsilon \le \varepsilon_{t0} \\ 1 & \varepsilon \le \varepsilon_{t0} \end{cases}.\qquad(4)$$

Additionally, we assume that the damage of mesoscopic element in multi-axial stress fields is also isotropic elastic. According to the method of extending one-

dimensional constitutive laws under uniaxial tensile stress to complex tensile stress conditions, which was proposed by Mazars and Pijaudier-Cabot[34] for a constitutive law of elastic damage, we can easily extend the constitutive law described above to a 3D stress state. Under multi-axial stress states, the element is still damaged in tensile mode when the equivalent maximum tensile strain, $\bar{\varepsilon}$, attains the above threshold strain $\varepsilon_{t0}$. Therefore, the constitutive law of an element that is subjected to multi-axial stresses can be easily obtained by substituting the strain $\varepsilon$ in Eqs. (3) and (4) with the equivalent principal strain $\bar{\varepsilon}$:

$$\bar{\varepsilon} = \sqrt{\langle-\varepsilon_1\rangle^2 + \langle-\varepsilon_2\rangle^2 + \langle-\varepsilon_3\rangle^2}, \tag{5}$$

where $\varepsilon_1$, $\varepsilon_2$, and $\varepsilon_3$ are the three principal strains, and $<>$ is a function defined as follows:

$$\langle x \rangle = \begin{cases} x & x \geq 0 \\ 0 & x < 0 \end{cases}. \tag{6}$$

It must be emphasized that the finite element analysis will halt if the modulus is set to zero. Therefore, a relatively small number, i.e. $10^{-5}$ is specified for the limit elastic modulus.

The above constitutive law only considers the situation when an element is damaged in the tensile failure mode. However, the compressive or shear failure mode also occurs when the elements are subjected to high compressive or shear stresses; thereafter, shear damage at the element scale level is also considered in our study for elements that are under compressive or shear stresses. The Mohr–Coulomb criterion, as expressed in Eq. (7), is selected to be the second damage threshold.

$$F = \sigma_1 - \frac{1+\sin\phi}{1-\sin\phi}\sigma_3 \geq f_{c0}. \tag{7}$$

where $\sigma_1$ and $\sigma_3$ are the maximum and minimum principal stresses respectively. $f_{c0}$ is the uniaxial compressive strength, and $\varphi$ is the internal friction angle of this element. We assume that $f_{c0}/f_{cr} = f_{t0}/f_{tr} = \lambda$ is true when the element is under uniaxial compression or tension.

With regard to the constitutive law in Supplementary Fig. 13 when the element is damaged in shear mode, the damage variable $D$ can be described as follows:

$$D = \begin{cases} 0 & \varepsilon < \varepsilon_{c0} \\ 1 - \frac{\lambda\varepsilon_{c0}}{\varepsilon} & \varepsilon \geq \varepsilon_{c0} \end{cases}, \tag{8}$$

where $\varepsilon_{c0}$ is the strain at the peak compressive principal stress, under uniaxial compressive stress state, which can be simply calculated as

$$\varepsilon_{c0} = \frac{f_{c0}}{E_0}. \tag{9}$$

The mechanical behavior of quasi-brittle materials under multi-axial compression is mainly characterized by a considerable increase in strength and pre-peak strain at a high confinement level. When an element is in the multi-axial stress state and its stress condition satisfies the Mohr–Coulomb criterion, shear damage occurs, and we must consider the effect of other principal stresses in this model during the damage evolution process.

When the Mohr–Coulomb criterion is satisfied, we can calculate the minimum principal strain (maximum compressive principal strain) $\varepsilon_{c0}$ at the peak value of maximum principal stress (maximum compressive principal stress)

$$\varepsilon_{c0} = \frac{1}{E_0}\left[f_{c0} + \frac{1+\sin\phi}{1-\sin\phi}\sigma_3 - \mu(\sigma_1 + \sigma_2)\right]. \tag{10}$$

In this respect, we assume that the shear damage evolution is only related to the maximum compressive principal strain $\varepsilon_1$. Hence, we use the maximum compressive principal strain $\varepsilon_1$ of the damaged element to substitute the uniaxial compressive strain in Eq. (8). Thus, the former Eq. (8) can be extended to triaxial stress states for shear damage.

$$D = \begin{cases} 0 & \varepsilon_1 < \varepsilon_{c0} \\ 1 - \frac{\lambda\varepsilon_{c0}}{\varepsilon_1} & \varepsilon_1 \geq \varepsilon_{c0} \end{cases}. \tag{11}$$

From the above expression of damage variable $D$, which is generally called the damage evolution law in damage mechanics, together with the Eq. (2), we can calculate the damaged elastic modulus of the element at each stress or strain level.

In this model, the element may gradually be damaged according to the above elastic damage constitutive law. Only elements whose ultimate tensile strain has been attained are displayed as fractures in black in the post-processing figures. Both tensile damage and shear damage lead to the mechanical property degradation of elements, but tensile strain is considered to be the direct cause of fracture initiation. In this respect, the initiation, propagation, and interaction of multiple fractures is easily simulated.

**Model validation**. As noted in the main text, because the integrated strength of the lithosphere (i.e., thickness multiplied by modulus) should generally control fracture initiation and growth, we are able to validate our approach by comparing results of a thin shell (20 km) model with a high elastic modulus (250,000 MPa) (Model 8) against results of a thick shell (200 km) model with a low elastic modulus (25,000 MPa) (Model 9) (Supplementary Fig. 1). The resultant similarity in fracture

spacing/polygon size is consistent with expectations (Supplementary Fig. 1). The range of elastic modulus values used is consistent with determinations from relationships between wave velocity and Earth's crust (e.g., refs. [35–39]).

A further question is the extent to which the results depend on the radial expansion rate, which is modeled as 0.03 km Myr$^{-1}$, or roughly what would be expected for a 50-km-thick lid. In nature, this warming timescale increases as lithospheric thickness squared, while the total uplift increases linearly with the thickness, thus the uplift rate is inversely proportional to the lithospheric thickness and would be half as much for a 100-km lid and twice as much for a 25-km lid. However, the expansion rate does not significantly impact results; the critical parameter is that the total strain reach about 1 km. We demonstrate this by expanding the reference model with rates twice greater and half of the rate in the rest of the models (specifically, 0.06 km and 0.015 km Myr$^{-1}$—see Supplementary Fig. 11). The different rates impact the development of the fracture network most significantly in terms of the timing of initial rupture. For the slow radial loading rate, wherein the amount of expansion during each time step is small (15 m at each step), the initiation rupture time of the sphere is later and the initial rapid rupture growth phase is smaller. For the rapid radial loading rate, wherein the amount of expansion during each time step is large (60 m at each step), the initiation rupture time of the sphere is earlier and the initial rapid rupture growth phase lasts longer. These differences occur because when the loading amount at each step is small (15 m at each step), the damage and failure of the element occur gradually, the energy is released gradually, and the stress is adjusted gradually. When the loading amount at each step is large (60 m at each step), the damage of the element increases sharply at a certain stage, the energy is released suddenly, and the stress is transferred. These differences notwithstanding, the general pattern holds across all models: initial rapid growth of a fracture network occurs at about 1 km of radial loading, followed by a slower phase of fracture network growth, regardless of lid thickness and expansion rate.

## Data availability
All modeling data generated in this study are available upon request.

## Code availability
The code RFPA3D (Rock Failure Process Analysis Code) was developed by lead author C.A.T., and is commercially available from Mechsoft Technology (Dalian) Co., Ltd.

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

## Acknowledgements

C.A.T. acknowledges support of the National Key Basic Research Development Plan (973) (No. 2014CB047100) and the Talent Cultivation Plan of "Xinghai Scholars" of Dalian University of Technology. A.A.G.W. acknowledges support from the Research Grants Council of Hong Kong, General Research Fund Grant #17305718.

## Author contributions

C.A.T. and A.A.G.W. formulated the research goals; C.A.T., Y.Y.W., T.H.M., and T.T.C. performed the modeling; and C.A.T., A.A.G.W., and W.B.M. wrote and revised the manuscript.

## Competing interests

The authors declare no competing interests.
