## [Peer Review File · Nature Communications]

Reviewers' comments:

Reviewer #1 (Remarks to the Author):

Breaking Earth's shell into a global plate network

This paper presents a mechanism by which plate tectonics was initiated on Earth. It is assumed that the Earth initially cooled very efficiently by heat piping. When heat piping is no longer effective, the interior heats up first and this heating causes an expansion and rupture of the lithosphere. The resulting fractures show a distribution of polygon structures whose spacing is dependent on lithospheric thickness and rheology. For these calculations a 3D spherical shell model (finite element code) was used which simulates the breaking of a brittle shell layer. The lower boundary condition is disturbed by a continuous displacement, which corresponds to heating and thus thermal expansion. The model does not show the initiation of plate tectonics but the initiation of a global network of rifts, which can then be important for the initiation of plate tectonics in interaction e.g. with lateral variations of the buoyancy forces in the lithosphere. This is an interesting study and can make an important new contribution to the general discussion on plate tectonics and its origin. However, some more detailed discussion and estimates are missing as to whether this mechanism can actually work.

All model calculations shown assume a fixed uplift rate of 0.03 km/Myr, which leads to an expansion of approx. 1.5 km in 50 Myr. To what extent do the results and the occurrence of the global network of rifts depend on this expansion rate?

Furthermore, it is estimated that this uplift requires a mean temperature increase of 500 K also in a time of 50 Myr. Is this realistic at all? This seems to be a lot, considering that this needs to be achieved mainly by radioactive decay after the phase of heat piping. This should be demonstrated.

I assume that this mechanism to form a global plate network is less efficient with heat pipe cooling for which the extrusion rate is not 100%. Is it possible to quantify?

During heating, the lithosphere becomes thinner. What impact will this have on the global network of rifts? The same applies if the uplift is not homogeneously distributed as e.g. a consequence of mantle flow.

Are the examples for Venus and Mars relevant here? As far as I know, the tectonic structures on Venus are quite young and can rather not be associated with processes after an early heat piping phase. The formation of the Valles Marineris is associated with the load of the volcanic region, Tharsis, and does not seem to me to be a good example either.

Reviewer #2 (Remarks to the Author):

Review of "Breaking Earth's shell into a global plate network" by C.A. Tang et al.

This paper presents 3D spherical shell models, an approach generally not used in geodynamics, in order to try to explain how the Earth's surface evolved from a single plate in its past, eventually breaking into several ones. The authors find that it is possible to break the lithosphere into a network of plates due to shallow processes, namely a fracture mechanism analogous to thermal expansion. The sizes of the plates originating from this process are dependent on the initial thickness and rheology of the lithosphere.

The question that this work addresses is pertinent and falls into a general community effort which tries to understand the evolution of the Earth into plate tectonics. The method is innovative, and it

is a good effort to try explaining the early Earth. In particular, I find it fitting that a different number of scenarios are explored, which include different lithosphere thicknesses. This study presents a new idea in order to explain the breaking of the Earth's shell into plates. From there, the authors speculate how can plate tectonics start. I believe it is suitable for publication in Nature Communications if the major (and minor) issues raised next are properly addressed.

Different major improvements are needed in the text to make this work clearer and more accurate. My main criticisms are:

As the authors point out in Line 28, the community hasn't converged on a typical plate scale. The authors miss the opportunity to discuss this more. I would suggest adding to the Discussion part of the paper a more detailed analysis of the implications of different lithospheric thicknesses to the plate sizes. There are some works that compute lithosphere thicknesses depending on other factors such as volcanism and intrusion rates (see for example Moore and Webb, Nature 2013, and Lourenço et al., 2018 Nat Geo, both cited in this work). One could elaborate on how more or less volcanism could impact plate size. And discuss what is the preferred thickness from the findings of the present work and why.

It would also be nice to justify the lithosphere thicknesses used (Line 73) with the same principle of thicknesses computed in other works.

The method is innovative and brings a new idea forward, but it does not take into account the influence of the mantle. For example, even if the lithosphere is broken into smaller plates, this does not mean that subduction will be active. As van Hunen and van den Berg (Lithos, 2008) point out, the mantle needs to cool down enough such that plates can subduct, instead of breaking-off due to a warm mantle. This needs to be discussed.

A big question that is not answered is when this could breaking happen. And mostly the timescales of it. It seems strange to me that the Earth is in a single plate for a potential long time (100s of Myrs to Gyr?) and then in about 40-60 Myrs the whole planet is divided in several plates... Is this reasonable? Is there any evidence for this? A better discussion is needed and linking with geological evidence would be great.

Lines 145-147 are inaccurate. The whole point about the Rozel et al, Nature 2017 paper is that high rates of intrusion are needed to explain geological evidence. Not high degrees of extrusion, meaning heat-pipes, as the text suggests.

Lines 147-148: This is also inaccurate. Heat-pipe cooling leads to a thick and cold crust and lithosphere, not the opposite... I also think that in the text that follows, a discussion about how high degrees of intrusion (that lead to a thin and warm lithosphere) would affect your interpretations.

The paragraph in lines 158-170 needs citations, otherwise it is purely speculative. Firstly, for previously published evidence of Mars, Venus, etc contractional features. Secondly, for when and how do people think these features happened; comparison with data is needed.

Detailed review:

Fig. 1: Only in lines 297-299 in the Methods it is said what the black lines in this (and similar figures) are. It would be important to write this in the main text.

Fig.2: Why does the line for total length of surface fractures keep increasing? What does that mean in practice?

Fig.3: Why are the lines for 20 and 40 km so different from the others? Also, why does 120 km lithosphere thickness grow faster than 60, 80 and 100 km?

Lines 33-35: Strange way to finish the sentence. Please rephrase.

Lines 44-48 and lines 53-55: In the first set of lines, the list uses (A), (B), etc, while afterwards (a), (b), etc are used.

Lines 56-57: "arbitrarily tune" sounds harsh. Even if geodynamical numerical models are definitely lacking something, we have learnt much from them, and they do take into account rock physics.

Lines 307-308: Can you put the elastic modulus values in perspective of what has been used before in other works?

Response-to-Reviewers:

This file contains the notes from the reviewer feedback, noted in purple text that is locally underlined to highlight key aspects, and our responses, noted in plain text.

Reviewers' comments:

Reviewer #1 (Remarks to the Author):

Breaking Earth's shell into a global plate network

This paper presents a mechanism by which plate tectonics was initiated on Earth. It is assumed that the Earth initially cooled very efficiently by heat piping. When heat piping is no longer effective, the interior heats up first and this heating causes an expansion and rupture of the lithosphere. The resulting fractures show a distribution of polygon structures whose spacing is dependent on lithospheric thickness and rheology. For these calculations a 3D spherical shell model (finite element code) was used which simulates the breaking of a brittle shell layer. The lower boundary condition is disturbed by a continuous displacement, which corresponds to heating and thus thermal expansion. The model does not show the initiation of plate tectonics but the initiation of a global network of rifts, which can then be important for the initiation of plate tectonics in interaction e.g. with lateral variations of the buoyancy forces in the lithosphere. This is an interesting study and can make an important new contribution to the general discussion on plate tectonics and its origin. However, some more detailed discussion and estimates are missing as to whether this mechanism can actually work.

All model calculations shown assume a fixed uplift rate of 0.03 km/Myr, which leads to an expansion of approx. 1.5 km in 50 Myr. To what extent do the results and the occurrence of the global network of rifts depend on this expansion rate?

Response to underlined text: we have included new text in the “Model Validation” section of the Methods (lines 325-342 of the revised manuscript), as well as a new figure in the Extended Data (Extended Data Fig. 7 of the revised manuscript) which directly address this issue. **In short, the expansion rate does not significantly impact results.** We demonstrate this (in the new materials) by expanding our reference model with different rates greater and lesser than the rate in our original modeling (specifically, 0.015 km/Myr and 0.06 km/Myr). The different rates impact the development of the fracture network largely in terms of the timing of initial rupture. For the slow radial loading rate, wherein the amount of expansion during each time step is small (15 m at each step), the initiation rupture time of the sphere is later and the initial rapid rupture growth phase is smaller. For the rapid radial loading rate, wherein the amount of expansion during each time step is large (60 m at each step), the initiation rupture time of the sphere is earlier and the initial rapid rupture growth phase lasts longer. These differences occur because when the loading amount at each step is small (15 m at each step), the damage and failure of the element occur gradually, the energy is released gradually, and the stress is adjusted gradually. When the loading amount at each step is large (60 m at each step), the damage of the element increases sharply at a certain stage, the energy is released suddenly, and the stress is transferred. These differences notwithstanding, the general pattern holds across all models: initial rapid growth of a fracture network occurs, followed by a slower phase of fracture network growth, regardless of lid thickness and expansion rate. The basic physical mechanism is simple and robust.

Furthermore, it is estimated that this uplift requires a mean temperature increase of 500 K also

in a time of 50 Myr. Is this realistic at all? This seems to be a lot, considering that this needs to be achieved mainly by radioactive decay after the phase of heat piping. This should be demonstrated.

Response to underlined text: There are two main issues here: the amount of temperature increase and the time for the temperature increase. The additions to the text explained in the comment-response directly above addresses the time for the temperature increase, demonstrating that the rate doesn't fundamentally change the overall system behavior. The overall temperature increase has been previously demonstrated by Moore and Webb (2013, Nature) as the difference between the heat-pipe and conduction geotherms. We explain these aspects more fully and with shorter, clearer sentences in the revised text, at lines 161-167. Finally, we note that radioactive decay is not the central issue here. There is no paucity of heat. The issue is that heating of the lithosphere via conduction of heat from below was dramatically suppressed by downwards advection of the lithosphere itself during heat-pipe cooling. As that suppression died away (as volcanic resurfacing and thus downwards advection waned) the lithosphere will rapidly warm towards the dramatically warmer conduction geotherm.

I assume that this mechanism to form a global plate network is less efficient with heat pipe cooling for which the extrusion rate is not 100%. Is it possible to quantify?

Response to underlined text: It is not clear that this assumption is valid. Intrusive magmatism delivers heat to the interior of the lid rather than the surface and therefore will alter the expansion caused by adjustment back to conductive equilibrium. Without a detailed model of this process, this will be difficult to quantify, but three points should be made. First, the rate of advection at the surface must be the same (otherwise the mantle is not getting the heat out) and there must consequently be a higher mantle temperature and melting rate (to make up for intrusive losses). Second, intrusion results in an accelerating downward advection through the lithosphere (intrusions push down the layers below) which also alters the temperature structure. An averaged solution to the temperature field is possible in this case, but it is not straightforward to evaluate. Third, there has been recent work on this question, namely by Rozel et al., 2017 Nature, which claims that the geological record supports significant intrusive rather than heat-pipe cooling, but these conclusions are overly optimistic. One argument is that the heat-pipe lithosphere would be too cold to produce observed tonalite-trondhjemite-granodiorite (TTG) intrusive rocks. While this is true in terms of average background temperatures, nonetheless in the heat-pipe model and in the geological record it is clear that TTG rocks are produced via transient, localized phenomena (i.e., these rocks show up in local pulse events, not continuous generation). Local pulses of heating during rapid local burial and/or delamination are indeed part of the heat-pipe model. A second argument from Rozel et al. is that they are able to tune their model to the Archean TTG rock record, in terms of which type of TTG is produced (i.e., low-, medium-, and high- pressure TTG). While it is true that they obtain a good fit to Archean TTG, they produce over half of their model TTG in the first 200 million years of the model evolution, roughly equivalent to the first half of the Eoarchean (~4.0-3.8 Ga). There is negligible material preserved from this period (and furthermore, strong potential that what is preserved is not representative), so their model-to-data comparison is apples to oranges. **In short, the whole concept of significant intrusive magmatism as an alternative to heat-pipe cooling is on shaky ground, and even if this did occur, it might not significantly alter the overall lithospheric thermal structure.** The active heat-pipe tectonics of Io demonstrate the basic viability of the heat-pipe cooling mechanism.

During heating, the lithosphere becomes thinner. What impact will this have on the global network of rifts?

Response to underlined text: Figure 3 clearly demonstrates that lid thickness variations from ~200 to ~20 km results in the same basic pattern, so it is reasonable for the first-order effort to assume that time-integrated thickness variations will produce second order effects at most. Such issues can be explored in later contributions for specialist audiences.

The same applies if the uplift is not homogeneously distributed as e.g. a consequence of mantle flow.

Response to underlined text: Dynamic topography can produce roughly a kilometer of uplift, but this is supported by bending in response to mantle tractions with negligible thermal expansion responses. Therefore this is likewise a second-order feature that can be explored in later contributions.

Are the examples for Venus and Mars relevant here? As far as I know, the tectonic structures on Venus are quite young and can rather not be associated with processes after an early heat piping phase. The formation of the Valles Marineris is associated with the load of the volcanic region, Tharsis, and does not seem to me to be a good example either.

Response to underlined text: We don't agree with some of these points and could argue over specifics here (e.g., for Venus see Moore et al., 2017 EPSL). Nonetheless as [1] many readers may have similar reactions, [2] this planetary section might therefore distract from our overall message, and [3] the editors specifically requested removal of this section, we have removed this paragraph (lines 158-170 in the original manuscript).

Reviewer #2 (Remarks to the Author):

Review of "Breaking Earth's shell into a global plate network" by C.A. Tang et al.

This paper presents 3D spherical shell models, an approach generally not used in geodynamics, in order to try to explain how the Earth's surface evolved from a single plate in its past, eventually breaking into several ones. The authors find that it is possible to break the lithosphere into a network of plates due to shallow processes, namely a fracture mechanism analogous to thermal expansion. The sizes of the plates originating from this process are dependent on the initial thickness and rheology of the lithosphere.

The question that this work addresses is pertinent and falls into a general community effort which tries to understand the evolution of the Earth into plate tectonics. The method is innovative, and it is a good effort to try explaining the early Earth. In particular, I find it fitting that a different number of scenarios are explored, which include different lithosphere thicknesses. This study presents a new idea in order to explain the breaking of the Earth's shell into plates. From there, the authors speculate how can plate tectonics start. I believe it is suitable for publication in Nature Communications if the major (and minor) issues raised next are properly addressed.

Different major improvements are needed in the text to make this work clearer and more accurate. My main criticisms are:

As the authors point out in Line 28, the community hasn't converged on a typical plate scale. The authors miss the opportunity to discuss this more. I would suggest adding to the Discussion part of the paper a more detailed analysis of the implications of different lithospheric thicknesses to the plate sizes.

Response to underlined text: We considered this approach, but we had decided that it would be too speculative with too few interpretative rewards. Nonetheless in response to the reviewer we add a minor comment on this now, at lines 174-177 of the revised manuscript, and we also added a figure illustrating how with increasing lid thickness, plate size increases (Extended Data Figure 6). Casual visual inspection of our figures, including the model ‘balls’ of Figure 3 and the many progressive evolutions shown in the Extended Data Figures, reveals that thicker lids yield larger plates, as made explicit in Extended Data Figure 6. However, it is questionable how relevant this result would be in terms of analyzing plate size beyond the initiation of plate tectonics. The models show generation of a rifted state with multiple plates. This state would be an immediate precursor to subduction tectonics. Nonetheless, because there is no role for subduction in influencing the plate sizes generated in our models, the model results can be argued to have limited importance for plate sizes during plate tectonic recycling. We reference the core literature on fracture spacing in layered materials (which our findings are consistent with), and we do want to explore this question further in future work, but speculation here should be kept limited.

There are some works that compute lithosphere thicknesses depending on other factors such as volcanism and intrusion rates (see for example Moore and Webb, Nature 2013, and Lourenço et al., 2018 Nat Geo, both cited in this work). One could elaborate on how more or less volcanism could impact plate size. And discuss what is the preferred thickness from the findings of the present work and why. It would also be nice to justify the lithosphere thicknesses used (Line 73) with the same principle of thicknesses computed in other works.

Response to underlined text: As noted above, the main result is that dramatic variations in lid thickness appear to have negligible impact on the physical mechanism for plate breaking introduced here (e.g., see Figure 3). We include lid thicknesses far thinner and thicker than almost any previously considered in such contexts (20 and 200 km) and still get the same basic outcomes. Therefore we consider it self-evident that there is no preferred thickness with respect to these models and the global plate breaking mechanism presented here. This may be an area for future study, with fracturing models applied to systems with rifting and subduction as opposed to plasticity (e.g., Mallard et al., 2016 Nature), but such work is well beyond the scope of the present work.

The method is innovative and brings a new idea forward, but it does not take into account the influence of the mantle. For example, even if the lithosphere is broken into smaller plates, this does not mean that subduction will be active. As van Hunen and van den Berg (Lithos, 2008) point out, the mantle needs to cool down enough such that plates can subduct, instead of breaking-off due to a warm mantle. This needs to be discussed.

Response to underlined text: The reviewer is primarily making a delamination argument, noting that in a hotter lithosphere, the bond between crust and mantle is weaker than the weak zones created by our process, so the dense material drips off leaving the near-surface low-density crust behind. However, the heat-pipe crust (and residuum) are not nearly as buoyant as modern basalt, being products of a much higher degree of melting, so they don’t resist subduction as effectively. Secondly, none of the models in van Hunen and van den Berg actually fail to subduct (and they acknowledge that subduction may continue if they run the models longer) — it may be drippy subduction, but material continues to descend. There is nothing particularly distinctive about their default choice of parameters — it is a (nearly) arbitrary reference, given our relative ignorance of material properties, deformation mechanisms, and composition. We ourselves don’t model the style of subduction, merely that (as van Hunen and van den Berg assume in how they set up their models!) there are weak zones available for subduction to take advantage of. Our modeling generates those weak zones. In this respect, our work is no different from contributions like Rey et al., 2014 Nature

or Gerya et al., 2015 Nature. The final point to make is that the mantle couldn't be so hot as to make a viable limit in this manner. If the mantle were heated up by the requisite ~300 degrees it would melt dramatically, and heat-pipe cooling itself would resume. Thus, volcanism is the limit preventing the excessively warm mantle problem, as already expounded in Moore and Webb (2013, Nature).

A big question that is not answered is when this could breaking happen. And mostly the timescales of it. It seems strange to me that the Earth is in a single plate for a potential long time (100s of Myrs to Gyr?) and then in about 40-60 Myrs the whole planet is divided in several plates... Is this reasonable? Is there any evidence for this? A better discussion is needed and linking with geological evidence would be great.

Response to underlined text: There are a variety of “subduction catastrophe” models, many of which are derived primarily from geological data such as the Barberton and Pilbara greenstone belt records, that involve similarly rapid transitions. In fact, models with gradual transitions to plate tectonics are relatively rare. We added a line to refer readers to a source on this subject, noting at lines 172-174 of the revised manuscript that the geological evidence is actually consistent with a rapid transition, on the timescale of the rapid growth of fractures in the modeling.

Lines 145-147 are inaccurate. The whole point about the Rozel et al, Nature 2017 paper is that high rates of intrusion are needed to explain geological evidence. Not high degrees of extrusion, meaning heat-pipes, as the text suggests.

Response to underlined text: We fully agree with the reviewer that this is the argument of the Rozel et al 2017 paper. This is why we cited it using the “cf.” term, which means “compare” and in our understanding is commonly used in the scientific literature to provide a citation that in fact disagrees with the premise being stated. If we are incorrect in this respect, we are happy to adjust the text. We generally feel that Rozel et al. are incorrect, for reasons enumerated above (see our response to Reviewer 1's question: I assume that this mechanism to form a global plate network is less efficient with heat pipe cooling for which the extrusion rate is not 100%. Is it possible to quantify?). This modeling-centered contribution does not appear to be the right venue to directly rebut them (a data-focused paper would be more appropriate). Therefore in this context we feel it is fair to the scientific community to acknowledge the differences via a “cf.” citation, and then proceed. If the reviewers or editors show that a different approach would be preferable, we are happy to make adjustments.

Lines 147-148: This is also inaccurate. Heat-pipe cooling leads to a thick and cold crust and lithosphere, not the opposite...

Response to underlined text: The reviewer is correct that heat-pipe cooling leads to a thick and cold lithosphere, but it is an intriguing difference vs. other models that heat-pipe lithosphere thins with time. It starts incredibly thick, but as the planet cools and volcanic advection wanes, so does the downwards advection of surface materials that create the thick lid. Conduction plays an increasing role in the lithospheric thermal structure, warming it and thinning the lithosphere from the base. We feel that this logic was reasonably well articulated in the original text (in original lines 147-157), but we endeavor to make it more clear in the revised manuscript by making the sentences shorter and adding in a little bit more redundancy in the description (see lines 152-167).

I also think that in the text that follows, a discussion about how high degrees of intrusion (that lead to a thin and warm lithosphere) would affect your interpretations.

Response to underlined text: As stated in the reply above (in response to Lines 145-147 are...), we feel that this is not necessarily an appropriate venue to rebut the Rozel et al arguments. Honestly we feel this is a potentially high-profile modeling-centric contribution, and as we feel the principal arguments against Rozel et al are based on very simple geological data and corresponding reasoning (see the response to Reviewer 1's question: I assume that this mechanism to form a global plate network...), it seems mean-spirited and not wholly relevant / necessary to include those arguments here. We are very willing to accept further guidance on this point.

The paragraph in lines 158-170 needs citations, otherwise it is purely speculative. Firstly, for previously published evidence of Mars, Venus, etc contractional features. Secondly, for when and how do people think these features happened; comparison with data is needed.

Response to underlined text: As stated above, we removed this section.

Detailed review:

Fig. 1: Only in lines 297-299 in the Methods it is said what the black lines in this (and similar figures) are. It would be important to write this in the main text.

Response to underlined text: We agree, we add this explanation to the Figure caption in the revised version (see lines 108-109).

Fig.2: Why does the line for total length of surface fractures keep increasing? What does that mean in practice?

Response to underlined text: The fractures keep growing after the rapid establishment of the global fracture network because the model Earth keeps expanding. In practice, this late continued growth isn't physically plausible nor relevant: thermal expansion can't go on forever, and in the geological environment it wouldn't be expected to continue. This is because if the fractures act as magma conduits, weighting lithosphere and initiating subduction (as in Gerya et al., 2015 Nature, for example), then the subduction process would effectively terminate the thermal expansion as an Earth surface concern. So, this is the flip side to modeling in which the initial conditions happen to be inconsequential (e.g., Moore and Webb, 2013 Nature) – here the final conditions are not consequential. This is discussed as part of lines 175-185 in the original manuscript, and lines 180-191 in the revised manuscript.

Fig.3: Why are the lines for 20 and 40 km so different from the others? Also, why does 120 km lithosphere thickness grow faster than 60, 80 and 100 km?

This was a function of modeling approach idiosyncrasies in terms of element numbers. We reran all of our models, from thin spherical shell models to thick spherical shell models, with the number of element layers increasing proportionally. The results essentially look the same, except that the seeming 40 to 60 km distinction is smoothed out. The new computational work took substantial time, which is the primary reason our resubmission took more than a month. Subtle changes to Figures 1, 2, 3, and Extended Data Figures 3, 4, 5, and the discussion of these figures resulted from this process (e.g., the same basic processes happen, but at ~30 Ma instead of ~28 Ma in some models).

Lines 33-35: Strange way to finish the sentence. Please rephrase.

Response to underlined text: We have edited this sentence in the revised version to read “Resultant fracture spacing is a function of lithospheric thickness and rheology, wherein geometrically-regular, polygonal-shaped tessellation is an energetically favored solution because it minimizes total crack length.” See lines 35-37 of the revised version.

Lines 44-48 and lines 53-55: In the first set of lines, the list uses (A), (B), etc, while afterwards (a), (b), etc are used.

Response to underlined text: We have edited this to be consistent in the revised text.

Lines 56-57: “arbitrarily tune” sounds harsh. Even if geodynamical numerical models are definitely lacking something, we have learnt much from them, and they do take into account rock physics.

Response to underlined text: We agree, we changed the text by removing the word “arbitrarily.”

Lines 307-308: Can you put the elastic modulus values in perspective of what has been used before in other works?

We added a note at lines 323-324 of the Materials and Methods section explaining how our values are consistent with the standard expectations from published explorations.

Reviewers' comments:

Reviewer #1 (Remarks to the Author):

The authors have answered most questions and comments satisfactorily. However, in my opinion there are still a few points open. For instance, they have shown that the expansion rate has no great influence on the results by varying the loading rate by factor of 2 smaller and larger. The main effect is that the time of fracturing is shifted depending on when the necessary radial loading of 1 is attained. It is therefore argued by the authors that the time scale of heating is not a critical point for their model. However, these tested values still represent extreme conditions, i.e. the lithosphere still has to heat up very strongly and very fast. What might be missing here is an information/ discussion as to when the effect is no longer sufficient to break the lithosphere. This means, when is the 1 km minimum radial load no longer reached (I assume this minimum radial value depends also on the mechanical parameters of the lithosphere)? There should be a minimum temperature increase, but also a maximum time for this temperature increase to obtain a global plate network. Without this information it is difficult to assess how relevant this effect is. And as far as the 500 K heating is concerned, reference is made to the Moore and Webb 2013 paper. But this study does not show how long it takes to heat the lithosphere by this value, only that the temperature profiles of a heat pipe lithosphere can deviate strongly from the conductive profile for 100% extrusion. Further, I disagree with the authors that intrusive magmatism is on shaky ground. The estimated extrusive/intrusive rate for terrestrial planets is about 1/5 or 1/10 (e.g. Crisp, 1984) One should rather ask oneself why this should have been different in early Earth. Heat piping is certainly effective in cooling the planetary interior but there's no reason to assume that heat piping requires 100% extrusion. Also for Io I'm not sure whether 100% extrusion is really generally accepted. (For instance, this Fall-AGU there was an abstract of Schools and Montesi showing that melt in the lithosphere of Io can't rise so easily). The model of Rozel et al. and its geological interpretation of the TTGs is in my opinion also rather secondary for this question. And if there is intrusive magmatism, the temperature profile in the lithosphere will go to higher temperatures - this can't be avoided, thus the effect of expansion will certainly be smaller. It is by no means secured that expansion would be enough then (see point above for a better analysis of the necessary temperature increase and the maximum time needed for this temperature increase to obtain a global plate network).

Reviewer #2 (Remarks to the Author):

Review of "Breaking Earth's shell into a global plate network" by C.A. Tang et al.

This paper presents 3D spherical shell models, an approach generally not used in geodynamics, in order to try to explain how the Earth's surface evolved from a single plate in its past, eventually breaking into several ones. The authors find that it is possible to break the lithosphere into a network of plates due to shallow processes, namely a fracture mechanism analogous to thermal expansion. The question that this work addresses is pertinent and falls into a general community effort which tries to understand the evolution of the Earth into plate tectonics.

The method is innovative in the Earth Sciences field and well explained in the Methods section. I find it fitting that a different number of scenarios are explored, which include different lithosphere thicknesses. I am happy about how the paper flows, and how it is easy to follow. I am also happy about how the authors rerun their simulations, and with most of the clarifications that the authors provided to the reviewers, especially by introducing explanations about uplift rates, geological background for the rapid development of fractures, and by removing the (more speculative) discussion about other terrestrial planets.

I believe it is suitable for publication in Nature Communications if the minor issues raised next are properly addressed.

My main criticism is that I still believe (and also reading the comments from the other reviewer) that a longer discussion on how intrusion would affect the results is needed. Even if the authors think that those works are on 'shaky ground' and too 'modeling-centered', I think that a substantial part of the community agrees with them, or at least has less reservations. Therefore, I believe that the issue should be discussed in the paper. It doesn't need to be an extensive discussion, but rather a few sentences discussing how would an intrusion-dominated Archean Earth affect the results, for example, how would different lid thicknesses and different lithosphere and mantle temperatures affect the results (the authors already discuss part of this in their answers to Reviewer #1).

Minor points:

Line 41: Heating due to what?

Fig.3: I would suggest making the colors for the lines for 20 and 200 km more distinct. Quite hard to distinguish them now.

Line 263: The italic in the second part of the sentence is not necessary.

Extended Fig.4: The numbers in the colorbar are barely visible in all of the figures. Better to make them bigger.

Response-to-Reviewers:

This file contains the notes from the reviewer feedback, noted in purple text that is locally underlined to highlight key aspects, and our responses, noted in plain text.

This is our second re-submission of this manuscript since it has been last sent to review, such that it is the 3rd re-submission overall. In our 2nd re-submission, we disagreed with the request from both reviewers to add additional text on the question of extrusive vs. intrusive magmatism to the main body of the manuscript. Now, we have acceded to this request. In the spirit of maintaining transparency for the open review process, we include the 2nd re-submission response-to-reviewer text at the end of this file, in *italics*. Much of the text is similar to the present response, with the principal difference being a more substantial exploration of the extrusive vs. intrusive question.

Prior to addressing the reviewer comments, **we provide a discussion on an edit that the reviewers did not explicitly request, but that is in line with the general thrust of their questions regarding model plausibility. Specifically, in our revised manuscript we highlight the maximum extent of thermal expansion**, which indicates that for our models, full development of a global fracture network is limited to lids with thicknesses of ~70 to 80 km and greater. Our parameterization is conservative, so it is likely that thinner lids would break similarly, but nonetheless we now articulate this limit in our revised materials. In text, we explain this limit at lines 157-167. In figures, we show the model only up to this limit in Figure 1 (stopping at 40 m.y., or 1.2 km, of expansion for the reference model), and we indicate where the models surpass the limit via dashed lines in Figures 2 and 3 and Extended Data Figures 5 and 7. In tables, in the caption of Extended Data Table 1 we note the simple calculation that provides the thermal expansion limit (standard textbook concepts, e.g., Turcotte and Schubert's 2002 Geodynamics book), and in a new right-side column of the table itself we provide the limit in km for each model shell.

Reviewers' comments:

Reviewer #1 (Remarks to the Author):

The authors have answered most questions and comments satisfactorily. However, in my opinion there are still a few points open.

For instance, they have shown that the expansion rate has no great influence on the results by varying the loading rate by factor of 2 smaller and larger. The main effect is that the time of fracturing is shifted depending on when the necessary radial loading of 1 is attained. It is therefore argued by the authors that the time scale of heating is not a critical point for their model. However, these tested values still represent extreme conditions, i.e. the lithosphere still has to heat up very strongly and very fast. What might be missing here is an information/discussion as to when the effect is no longer sufficient to break the lithosphere. This means, when is the 1 km minimum radial load no longer reached (I assume this minimum radial value depends also on the mechanical parameters of the lithosphere)? There should be a minimum temperature increase, but also a maximum time for this temperature increase to obtain a global plate network. Without this information it is difficult to assess how relevant this effect is. And as far as the 500 K heating is concerned, reference is made to the Moore and Webb 2013 paper. But this study does not show how long it takes to heat the lithosphere by this value, only that the temperature profiles of a heat pipe lithosphere can deviate strongly from the conductive profile for 100% extrusion.

Response to underlined text / the above paragraph: The time it takes to heat the lithosphere by conduction can be estimated from a conductive timescale, which depends on the thickness of the lithosphere and the thermal diffusivity (e.g., see the Turcotte and Schubert 2002 Geodynamics textbook). Although rough, this provides a reasonable estimate for the time it takes for significant changes to the lithospheric temperature structure (about 30 Myr for a 50 km thick lid). This is discussed in the revised text, at lines 155-161.

Further, I disagree with the authors that intrusive magmatism is on shaky ground. The estimated extrusive/intrusive rate for terrestrial planets is about 1/5 or 1/10 (e.g. Crisp, 1984). One should rather ask oneself why this should have been different in early Earth. Heat piping is certainly effective in cooling the planetary interior but there's no reason to assume that heat piping requires 100% extrusion. Also for Io I'm not sure whether 100% extrusion is really generally accepted. (For instance, this Fall-AGU there was an abstract of Schools and Montesi showing that melt in the lithosphere of Io can't rise so easily). The model of Rozel et al. and its geological interpretation of the TTGs is in my opinion also rather secondary for this question. And if there is intrusive magmatism, the temperature profile in the lithosphere will go to higher temperatures - this can't be avoided, thus the effect of expansion will certainly be smaller. It is by no means secured that expansion would be enough then (see point above for a better analysis of the necessary temperature increase and the maximum time needed for this temperature increase to obtain a global plate network).

Response to the above paragraph, and to the similar paragraph from Reviewer 2 (below): Many aspects of this discussion are included in the prior response to reviewers. Nonetheless, there are further arguments for readers and the reviewers to consider, in support of our treatment of the extrusive / intrusive question:

The idea that the estimated extrusive/intrusive proportions for terrestrial planets is 1/5 to 1/10 is itself “shaky ground.” This proportionality has received very little detailed study, as emphasized by Crisp herself (of the Crisp, 1984, *J Volcanology Geotherm Res: Rates of magma emplacement and volcanic output* paper). In a follow-up paper which explicitly attempts to re-do the Crisp (1984) analysis with the benefit of twenty years' worth of additional data – namely: White, Crisp, and Spera, 2006, *G-cubed: Long-term volumetric eruption rates and magma budgets* – the authors emphasize the uncertainty and lack of study. They state that they “proceed with an analysis if for no other reason than to highlight that this issue has received so little attention” (quote from the end of their paragraph [15]).

There are basic limitations in the derivation of the 1/5 to 1/10 extrusive/intrusive ratios, as well-explained by Crisp (1984) and White, Crisp, Spera (2006). Foremost, they note that in the places where good exposure permits robust estimates of the volume of volcanic materials, the intrusives are buried from view, whereas in the places where the intrusives are well-exposed, the volcanics are largely eroded away. Neither contribution (nor any paper citing them that we found) attempted to estimate volcanic volumes with the benefit of adjacent sediment volumes hosting eroded igneous material, which itself would be a similarly challenging exercise that should tilt the ratios towards greater extrusive volumes. There are a variety of other approaches mentioned by these authors (e.g., seismic and gravity measurements of deep intrusives), many of which they incorporate in their surveys, and each with their own challenges. The resultant ratios vary a lot. At the extreme end, ratios of 100/1 or even 200/1 in favor of volcanics have been determined for the Coso volcanic field in California (Table 3 of White et al., 2006).

A clear issue with the way the 1/5 to 1/10 extrusive/intrusive ratios have been used in the recent community discussion on heat-pipe volcanism is the assumption that such ratios can be used as an “estimated extrusive/intrusive rate for terrestrial planets” regardless of magmatic system setting. The Phanerozoic magmatic systems that are widely considered most similar to Archaean volcanic systems are basaltic hotspot volcanic systems like the Hawaiian island chain and/or basaltic large igneous provinces (LIPs). **It is therefore worth further consideration of what data the 1/5 to 1/10 extrusive/intrusive ratios comes from, and the specific extrusive/intrusive ratios that apply to Hawaii and LIPs.** The following are the salient take-aways:

1. The 1/5 to 1/10 ratio is determined mostly from plate margin systems, with White et al. (2006) preferring the 1/5 ratio but acknowledging great variability.
2. White et al. (2006) demonstrate that oceanic hotspots and LIPs have substantially higher volcanic extrusion rates (by one and two orders of magnitude, respectively) and extrusive/intrusive ratios vs. all other Earth systems
3. Both Crisp (1984) and White et al. (2006) favor a 1/5 ratio for Hawaii, but later gravity work demonstrates a 9/1 to >7/3 ratio (Flinders et al., 2013, *GRL: Intrusive dike complexes, cumulate cores, and the extrusive growth of Hawaiian volcanoes*).
4. Crisp’s (1984) 1/5 ratio estimate for oceanic plateau (i.e., oceanic LIPs) is based primarily on her Hawaii estimate (see point 3 immediately above).
5. Both Crisp (1984) and White et al. (2006) acknowledge that basaltic volcanic systems have larger ratios than other systems, noting that “well-known basaltic shields do have” extrusive/intrusive ratios of 1/1 to 1/2.
6. Other literature suggests that LIPs have roughly 1/1 or larger ratios (e.g., Ridley and Richards, 2010, *G-cubed: Deep crustal structure beneath large igneous provinces and the petrologic evolution of flood basalts*), with the intrusions occurring largely as underplating (which would be recycled quickly into the convecting mantle in a downwards advecting heat-pipe system and would not substantially impact the overlying lithospheric thermal structure).

Further consideration of Io, the only active heat-pipe system in our solar system, is likewise useful. Io presently manages to move over a TeraWatt of heat through a lithosphere that supports 20 km - high mountains. This suggests strongly that intrusion is not significantly weakening the lithosphere, nor is magma having any difficulty rising to the surface. (Indeed, when the NASA New Horizons probe whipped by Io in 2007, it snapped photos of active volcanism. The .gif file on this page [<https://www.nasa.gov/topics/solarsystem/features/io-volcanoes-displaced.html>] is a beautiful illustration, showing an active volcanic plume mid-eruption via a series of five images taken over eight minutes).

Io’s volcanic resurfacing rate is 1 cm/yr, such that its ~50 km thick lithosphere would be entirely recycled in ~5 million years by volcanic material alone. Total recycling would be accelerated if intrusive additions are volumetrically significant. Because the amount of material erupting at the surface is set by the amount of heat being produced in the interior, additional material intruded in the lid accelerates the downward advection deeper in the lid. This accelerating advection term reduces the ability of intruded heat to alter the shape of the geotherm as intruded material and heat are pushed downward more and more rapidly. The implications of this for subsequent expansion are not clear, since a thicker lithosphere would result, allowing smaller temperature changes to integrate to the same volume change, but a self-consistent model of this process is lacking and beyond the scope of the present paper.

All this said, we do not discount intrusive emplacement, but at present our ability to quantify it is strongly limited, and thus we choose the simplest end member.

To express some fraction of the preceding concepts in the main text, we changed the text at what is now line 147, and the following lines. These formerly appeared as follows (with references written out):

“Models with dominantly volcanic cooling - termed heat-pipe cooling - are consistent with the early Earth’s geology (Moore and Webb, 2013, cf. Rozel et al., 2017) as well as the preserved lithospheres of the solar system’s other terrestrial planetary bodies (Moore et al., 2017).”

Now, we replaced this with the following text (again, with references written out):

“Models with dominantly volcanic cooling - termed heat-pipe cooling - are consistent with the early Earth’s geology (Moore and Webb, 2013) as well as the preserved lithospheres of the solar system’s other terrestrial planetary bodies (Moore et al., 2017). Recently, some workers (Rozel et al., 2017; Lourenco et al., 2018) have argued that intrusive magmatism should dominate early cooling. This argument is largely based on early work (Crisp, 1984) which found that Earth’s Phanerozoic magmatic systems generally display 1:5 to 1:10 extrusive-to-intrusive ratios. However, such findings were based on limited constraints and on primarily plate tectonic magmatic systems. Although constraints remain limited and substantial variability in extrusive-to-intrusive ratios exists, subsequent work has shown that the basaltic hotspot and basaltic large igneous province systems – which are widely accepted as our closest Phanerozoic Earth analogues to the major Archean magmatic systems – have far greater ratios vs. all other systems, generally at 1:1 or even 9:1 (White et al., 2006; Ridley and Richards, 2010; Flinders et al., 2013). [Further exploration of this issue can be found in the rounds of reviews and responses to reviewers, published alongside this paper].”

Reviewer #2 (Remarks to the Author):

Review of “Breaking Earth’s shell into a global plate network” by C.A. Tang et al.

This paper presents 3D spherical shell models, an approach generally not used in geodynamics, in order to try to explain how the Earth’s surface evolved from a single plate in its past, eventually breaking into several ones. The authors find that it is possible to break the lithosphere into a network of plates due to shallow processes, namely a fracture mechanism analogous to thermal expansion. The question that this work addresses is pertinent and falls into a general community effort which tries to understand the evolution of the Earth into plate tectonics.

The method is innovative in the Earth Sciences field and well explained in the Methods section. I find it fitting that a different number of scenarios are explored, which include different lithosphere thicknesses. I am happy about how the paper flows, and how it is easy to follow. I am also happy about how the authors rerun their simulations, and with most of the clarifications that the authors provided to the reviewers, especially by introducing explanations about uplift rates, geological background for the rapid development of fractures, and by removing the (more speculative) discussion about other terrestrial planets.

I believe it is suitable for publication in Nature Communications if the minor issues raised next are properly addressed.

My main criticism is that I still believe (and also reading the comments from the other reviewer) that a longer discussion on how intrusion would affect the results is needed. Even if the authors think that those works are on ‘shaky ground’ and too ‘modeling-centered’, I think that a substantial part of the community agrees with them, or at least has less reservations. Therefore, I believe that the issue should be discussed in the paper. It doesn’t need to be an extensive discussion, but rather a few sentences discussing how would an intrusion-dominated Archean Earth affect the results, for example, how would different lid thicknesses and different lithosphere and mantle temperatures affect the results (the authors already discuss part of this in their answers to Reviewer #1).

A response to this prior paragraph is provided above where we respond to similar feedback from Reviewer 1 (the link is noted as such in bold text).

Minor points:

Line 41: Heating due to what?

Response (this is now line 38 – “anticipated warming of the early lithosphere”): The lithospheric warming occurs with the cessation of heat-pipe cooling. As an edit, we now provide a reference link to the relevant citation here. The issue is discussed at length in the main text, but additional discussion here would over-lengthen and over-complicate the bold text.

Fig.3: I would suggest making the colors for the lines for 20 and 200 km more distinct. Quite hard to distinguish them now.

Response: We edited this accordingly, thanks for the suggestion.

Line 263: The italic in the second part of the sentence is not necessary.

Response: Good catch! We edited this accordingly.

Extended Fig.4: The numbers in the colorbar are barely visible in all of the figures. Better to make them bigger.

Response: Thanks, we edited this accordingly.

We made one more minor edit: we changed the sign convention when showing displacement, such that movement to the right is now positive in Figures 1, 3, and Extended Data Figure 4. This change was made to be visually more similar to a Cartesian plot.

Response-to-Reviewers (2nd re-submission):

This file contains the notes from the reviewer feedback, noted in purple text that is locally underlined to highlight key aspects, and our responses, noted in plain text.

Prior to addressing the reviewer comments, we provide a discussion on an edit that the reviewers did not explicitly request, but that is in line with the general thrust of their questions regarding model plausibility. Specifically, in our revised manuscript we highlight the maximum extent of thermal expansion, which indicates that for our models, full development of a global fracture network is limited to lids with thicknesses of ~70 to 80 km and greater. Our parameterization is conservative, so it is likely that thinner lids would break similarly, but nonetheless we now articulate this limit in our revised materials. In text, we explain this limit at lines 157-167. In figures, we show the model only up to this limit in

Figure 1 (stopping at 40 m.y., or 1.2 km, of expansion for the reference model), and we indicate where the models surpass the limit via dashed lines in Figures 2 and 3 and Extended Data Figures 5 and 7. In tables, in the caption of Extended Data Table 1 we note the simple calculation that provides the thermal expansion limit (standard textbook concepts, e.g., Turcotte and Schubert's 2002 Geodynamics book), and in a new right-side column of the table itself we provide the limit in km for each model shell.

Reviewers' comments:

Reviewer #1 (Remarks to the Author):

The authors have answered most questions and comments satisfactorily. However, in my opinion there are still a few points open.

For instance, they have shown that the expansion rate has no great influence on the results by varying the loading rate by factor of 2 smaller and larger. The main effect is that the time of fracturing is shifted depending on when the necessary radial loading of 1 is attained. It is therefore argued by the authors that the time scale of heating is not a critical point for their model. However, these tested values still represent extreme conditions, i.e. the lithosphere still has to heat up very strongly and very fast. What might be missing here is an information/discussion as to when the effect is no longer sufficient to break the lithosphere. This means, when is the 1 km minimum radial load no longer reached (I assume this minimum radial value depends also on the mechanical parameters of the lithosphere)? There should be a minimum temperature increase, but also a maximum time for this temperature increase to obtain a global plate network. Without this information it is difficult to assess how relevant this effect is. And as far as the 500 K heating is concerned, reference is made to the Moore and Webb 2013 paper. But this study does not show how long it takes to heat the lithosphere by this value, only that the temperature profiles of a heat pipe lithosphere can deviate strongly from the conductive profile for 100% extrusion.

Response to underlined text / the above paragraph: *The time it takes to heat the lithosphere by conduction can be estimated from a conductive timescale, which depends on the thickness of the lithosphere and the thermal diffusivity (e.g., see the Turcotte and Schubert 2002 Geodynamics textbook). Although rough, this provides a reasonable estimate for the time it takes for significant changes to the lithospheric temperature structure (about 30 Myr for a 50 km thick lid). This is discussed in the revised text, at lines 155-161.*

Further, I disagree with the authors that intrusive magmatism is on shaky ground. The estimated extrusive/intrusive rate for terrestrial planets is about 1/5 or 1/10 (e.g. Crisp, 1984) One should rather ask oneself why this should have been different in early Earth. Heat piping is certainly effective in cooling the planetary interior but there's no reason to assume that heat piping requires 100% extrusion. Also for Io I'm not sure whether 100% extrusion is really generally accepted. (For instance, this Fall-AGU there was an abstract of Schools and Montesi showing that melt in the lithosphere of Io can't rise so easily). The model of Rozel et al. and its geological interpretation of the TTGs is in my opinion also rather secondary for this question. And if there is intrusive magmatism, the temperature profile in the lithosphere will go to higher temperatures - this can't be avoided, thus the effect of expansion will certainly be smaller. It is by no means secured that expansion would be enough then (see point above for a better analysis of the necessary temperature increase and the maximum time needed for this temperature increase to obtain a global plate network).

Response to the above paragraph, and to the similar paragraph from Reviewer 2 (below):

Many aspects of this discussion are included in the prior response to reviewer, e.g., the disconnect between the data used to establish amounts of different types of TTG created (heavily weighted to the late Archean) and the Rozel et al. model generation of different types of TTG (heavily weighted to the model early Archean). As these reviews and responses will apparently be publicly available, there is already a lot for the interested reader to consider in this respect. Even though we clearly disagree on the science, we initially included the citation in the main text, without reviewer prompting, because other folks might want to consider it / find it relevant. So, we feel that at this point, there's a lot of spilt ink already, and further modifications to the main text are unwarranted.

Nonetheless, there are further arguments for readers and the reviewers to consider, in support of our treatment of the extrusive / intrusive question:

We do not discount intrusive emplacement, we simply have no way to quantify it (nor does anyone else) and thus we choose the simplest end member. We should emphasize that Io presently manages to move over a TeraWatt of heat through a lithosphere that supports 20 km - high mountains. This suggests strongly that intrusion is not significantly weakening the lithosphere, nor is magma having any difficulty rising to the surface. Another point to make clear is that, since the amount of material erupting at the surface is set by the amount of heat being produced in the interior, additional material intruded in the lid accelerates the downward advection deeper in the lid. This accelerating advection term reduces the ability of intruded heat to alter the shape of the geotherm as intruded material and heat are pushed downward more and more rapidly. The implications of this for subsequent expansion are not clear, since a thicker lithosphere would result, allowing smaller temperature changes to integrate to the same volume change, but a self-consistent model of this process is lacking and beyond the scope of the present paper.

Reviewer #2 (Remarks to the Author):

Review of "Breaking Earth's shell into a global plate network" by C.A. Tang et al.

This paper presents 3D spherical shell models, an approach generally not used in geodynamics, in order to try to explain how the Earth's surface evolved from a single plate in its past, eventually breaking into several ones. The authors find that it is possible to break the lithosphere into a network of plates due to shallow processes, namely a fracture mechanism analogous to thermal expansion. The question that this work addresses is pertinent and falls into a general community effort which tries to understand the evolution of the Earth into plate tectonics.

The method is innovative in the Earth Sciences field and well explained in the Methods section. I find it fitting that a different number of scenarios are explored, which include different lithosphere thicknesses. I am happy about how the paper flows, and how it is easy to follow. I am also happy about how the authors rerun their simulations, and with most of the clarifications that the authors provided to the reviewers, especially by introducing explanations about uplift rates, geological background for the rapid development of fractures, and by removing the (more speculative) discussion about other terrestrial planets.

I believe it is suitable for publication in Nature Communications if the minor issues raised next are properly addressed.

My main criticism is that I still believe (and also reading the comments from the other reviewer) that a longer discussion on how intrusion would affect the results is needed. Even if the authors think that those works are on ‘shaky ground’ and too ‘modeling-centered’, I think that a substantial part of the community agrees with them, or at least has less reservations. Therefore, I believe that the issue should be discussed in the paper. It doesn’t need to be an extensive discussion, but rather a few sentences discussing how would an intrusion-dominated Archean Earth affect the results, for example, how would different lid thicknesses and different lithosphere and mantle temperatures affect the results (the authors already discuss part of this in their answers to Reviewer #1).

A response to this prior paragraph is provided above where we respond to similar feedback from Reviewer 1 (the link is noted as such in bold text).

Minor points:

Line 41: Heating due to what?

Response (this is now line 38 – “anticipated warming of the early lithosphere”): *The lithospheric warming occurs with the cessation of heat-pipe cooling. As an edit, we now provide a reference link to the relevant citation here. The issue is discussed at length in the main text, but additional discussion here would over-lengthen and over-complicate the bold text.*

Fig.3: I would suggest making the colors for the lines for 20 and 200 km more distinct. Quite hard to distinguish them now.

Response: *We edited this accordingly, thanks for the suggestion.*

Line 263: The italic in the second part of the sentence is not necessary.

Response: *Good catch! We edited this accordingly.*

Extended Fig.4: The numbers in the colorbar are barely visible in all of the figures. Better to make them bigger.

Response: *Thanks, we edited this accordingly.*

We made one more minor edit: *we changed the sign convention when showing displacement, such that movement to the right is now positive in Figures 1, 3, and Extended Data Figure 4. This change was made to be visually more similar to a Cartesian plot.*

REVIEWERS' COMMENTS:

Reviewer #1 (Remarks to the Author):

I think that the paper can now be published and this work is certainly interesting to stimulate further discussion on the proposed mechanism of plate tectonics initiation. Nevertheless, I would like to raise two points again, which have already been mentioned in my previous reviews - even if it may seem nitpicking, but I have a feeling that important aspects are still missing in the discussion.

1) The authors have now explained in detail why they think that their end-ember model (full extrusion) is a reasonable assumption. I appreciate that they have added the arguments, but they still haven't said anything about what it means for their model if the relative extrusion rate looks different - which we cannot exclude at this point. It remains to be assumed that breaking is then no longer so easy, even with a small amount of intrusions.

2) The authors have now estimated how long it takes for a lithosphere to heat up so that the mechanism of global rifting by thermal expansion works and added the following sentences: "That is, warming of 50 km of lithosphere by an average of 500 degrees (half the difference between surface and interior temperatures) produces an uplift of 0.75 km (multiplying lid thickness by the thermal expansion coefficient and the temperature change). The thermal conduction timescale for a 50 km lid is about 30 Myr yielding an expansion rate similar to our assumed 0.03 km/Myr."

They use an example of 50 km although they argue: "a global rift network develops across lithospheres with thicknesses of 80 km and greater". Doing the same exercise with 100 km lid thickness, the time to heat the lid is about ~120 Myr and for 150 km it is even 270 Myr. Thus, heating may be slower than the time when the critical state of fracturing is reached (see Fig.2 and 3). The time scales for the thicker lids are also lower than those shown in Extended data Fig. 7. Does the mechanism still work if the time scale of heating is very slow or does this mean that the lid must not be too thick either?

Reviewer #2 (Remarks to the Author):

The authors have answered my questions and comments in a satisfactory way. I highlight the clear work that was put into discussing the intrusion vs extrusion ratios on Earth based on previously published work. I believe the paper is good to go, especially as readers will have access to the reviews and answers, together with the paper. I leave one very minor issue, I believe that in the new text added (line 150) it is better to use 'researchers' instead of 'workers', especially because 'work' shows up in the next sentence.

Response-to-Reviewers:

This file contains the notes from the reviewer feedback, noted in purple text that is locally underlined to highlight key aspects, and our responses, noted in plain text.

Prior to addressing the reviewer comments, we note that for publication, we had to remove references from the Abstract paragraph, as such inclusions are not consistent with Nature Communications' publication style. This directly impacts how we addressed a question from Reviewer 2 in a prior review round, who had noted that in the Abstract we had not explained anticipated warming of the early lithosphere. We noted that this was in fact a prior result (and it would take substantial text to explain it in the Abstract, although it is certainly explained in the main text), so our solution was to cite the prior work that explained this feature. However, with the removal of references, that solution no longer works, so now we have rephrased the sentence as follows: "Therefore, warming of the early lithosphere itself – as anticipated by previous studies – should lead to failure, propagating fractures, and the conditions necessary for the onset of multi-plate tectonics."

Reviewers' comments:

Reviewer #1 (Remarks to the Author):

I think that the paper can now be published and this work is certainly interesting to stimulate further discussion on the proposed mechanism of plate tectonics initiation. Nevertheless, I would like to raise two points again, which have already been mentioned in my previous reviews - even if it may seem nitpicking, but I have a feeling that important aspects are still missing in the discussion.

1) The authors have now explained in detail why they think that their end-ember model (full extrusion) is a reasonable assumption. I appreciate that they have added the arguments, but they still haven't said anything about what it means for their model if the relative extrusion rate looks different - which we cannot exclude at this point. It remains to be assumed that breaking is then no longer so easy, even with a small amount of intrusions.

Response to underlined text / the above paragraph: Without doing a second study, this cannot be directly addressed, but nonetheless we added some perspective on it. In lines 194-201 of the revised text (i.e., the end of the fourth paragraph in the Discussion and Conclusions section), we added the following text:

"This assumes that intrusion has not significantly warmed and softened the lid (cf. refs. ^{21, 22}), which would reduce thermal expansion following the end of heat-pipe volcanism and thus potentially limit the development of a fracture network. While the actual ratio of intrusion to extrusion is not well-constrained for any planetary system (both Crisp²⁶ and White et al.²⁸ are clear on this point), nor are the processes that control this ratio, it is nonetheless evident that the lithosphere of Io – our solar system's only active heat-pipe body – is not softened by high rates of intrusion, given its support of large topographic features.³¹"

This new text adds more explanation for why this may well be a moot point (via comparison with Io), and it explicitly acknowledges that negligible impacts from intrusion is a built-in assumption of this work.

2) The authors have now estimated how long it takes for a lithosphere to heat up so that the mechanism of global rifting by thermal expansion works and added the following sentences: “That is, warming of 50 km of lithosphere by an average of 500 degrees (half the difference between surface and interior temperatures) produces an uplift of 0.75 km (multiplying lid thickness by the thermal expansion coefficient and the temperature change). The thermal conduction timescale for a 50 km lid is about 30 Myr yielding an expansion rate similar to our assumed 0.03 km/Myr.”

They use an example of 50 km although they argue: “a global rift network develops across lithospheres with thicknesses of 80 km and greater”. Doing the same exercise with 100 km lid thickness, the time to heat the lid is about ~120 Myr and for 150 km it is even 270 Myr. Thus, heating may be slower than the time when the critical state of fracturing is reached (see Fig.2 and 3). The time scales for the thicker lids are also lower than those shown in Extended data Fig. 7. Does the mechanism still work if the time scale of heating is very slow or does this mean that the lid must not be too thick either?

Response to underlined text / the above paragraph: The experiments shown in the Supplementary Information (Supplementary Figure 11) using different expansion rates demonstrate that the development of the global fracture network depends on the total strain and not the strain rate for rates within an order of magnitude of our benchmark value. For simplicity, we use a constant rate and note that the critical parameter is that the total strain reach about 1km. We revised the text to clarify this by (1) adding clarifying text on this point in lines 347-351 and 366 of the Model Validation section of the Methods (see text quoted below) and (2) referring interested readers to this material via a parenthetical phrase in line 188-199 of the main text. Here are the edits at 327-351 and 366 (with *pre-existing text in italics*):

“A further question is the extent to which the results depend on the radial expansion rate, which is modeled as 0.03 km Myr⁻¹, or roughly what would be expected for a 50 km thick lid. In nature, this warming timescale increases as lithospheric thickness squared, while the total uplift increases linearly with the thickness, thus the uplift rate is inversely proportional to the lithospheric thickness and would be half as much for a 100 km lid and twice as much for a 25 km lid. However, the expansion rate does not significantly impact results; the critical parameter is that the total strain reach about 1 km. We demonstrate this by...”

“...initial rapid growth of a fracture network occurs at about 1 km of radial loading, followed by...”

Reviewer #2 (Remarks to the Author):

The authors have answered my questions and comments in a satisfactory way. I highlight the clear work that was put into discussing the intrusion vs extrusion ratios on Earth based on previously published work. I believe the paper is good to go, especially as readers will have access to the reviews and answers, together with the paper. I leave one very minor issue, I believe that in the new text added (line 150) it is better to use ‘researchers’ instead of ‘workers’, especially because ‘work’ shows up in the next sentence.

Response to underlined text / the above paragraph: Thanks for your kind words about the intrusion vs extrusion ratios work, and for your rephrasing advice, which we have followed.